# Analysis of Maternal Positions during the Dilation and Expulsive Phase and Their Relationship with Perineal Injuries in Eutocic Deliveries Attended by Midwives

**DOI:** 10.3390/healthcare12040441

**Published:** 2024-02-08

**Authors:** Cristian Martín-Vázquez, Noelia Goás-Gómez, Natalia Calvo-Ayuso, Laura Rosón-Matilla, Enedina Quiroga-Sánchez, Rubén García-Fernández

**Affiliations:** 1Department of Nursing and Physiotherapy, Campus de Ponferrada, Universidad de León, 24401 León, Spain; cmartv@unileon.es; 2Centro de Salud Vilalba, Servizo Galego de Saúde (SERGAS), 27800 Lugo, Spain; noeliagoasgomez@gmail.com; 3SALBIS Research Group, Department of Nursing and Physiotherapy, Faculty of Health Sciences, Campus de Ponferrada, Universidad de León, 24401 León, Spain; equis@unileon.es (E.Q.-S.); rgarcf@unileon.es (R.G.-F.); 4Hospital El Bierzo, 24404 Ponferrada, Spain; laurarosonmatilla@hotmail.com; 5Nursing Research, Innovation and Development Centre of Lisbon (CIDNUR), Nursing School of Lisbon, 1600-190 Lisbon, Portugal

**Keywords:** second labor stage, parturition, posture, episiotomy, midwife

## Abstract

This cross-sectional descriptive study aimed to analyze the relationship between maternal positions during the expulsion phase and perineal outcomes in 367 eutocic births attended by midwives or midwifery residents at a public hospital in northern Spain in 2018. A total of 94.3% of women opted for horizontal positions. Limited sacral retroversion was observed in 71.7%, potentially influencing perineal outcomes. A low incidence of tears indicated effective management during the expulsive phase, with an episiotomy rate of 15.3%, which was slightly above the 15% standard. Primiparity and maternal age were identified as risk factors associated with episiotomy. Additionally, sacral mobilization and vertical positions during delivery were significantly related to fewer perineal injuries, suggesting benefits for both mother and newborn. The correlation between maternal positions and the need for epidural analgesia highlighted the importance of considering these in pain management during childbirth. Despite limitations, the study provides valuable insight into obstetric practices and advocates for a woman-centered approach that respects autonomy during childbirth. Further research is needed to explore biomechanical parameters and enhance childbirth experiences.

## 1. Introduction

Postural changes and autonomy of movement throughout the birth process, especially in the second stage of labor, are crucial to its progression and influence the immediate, short-, and long-term outcomes for both the woman and the newborn. Throughout history, the postures adopted by women during labor have undergone significant changes [1]. Historically, upright postures, such as standing, sitting, squatting, or kneeling, were preferred because of their emphasis on freedom of movement and autonomy, allowing women to choose the posture that best suited them at any given moment. However, from the 17th century onwards, there was a marked shift toward the predominant use of the horizontal position during childbirth. This shift was aimed at increasing the comfort of the professionals involved, at the expense of the woman’s autonomy and control over her own birthing process [2].

In this context, episiotomy became popular in the 17th century. This procedure consists of a deliberate cut in the perineum to widen the vaginal opening during childbirth, affecting the skin, mucosa, and muscles. Initially introduced to prevent perineal tears and treat fetal complications such as fetal hypoxia and shoulder dystocia, its current indication is limited to situations of suspected fetal distress [3].

Midwives play a crucial role in childbirth care, significantly contributing to the improvement of maternal and neonatal outcomes. Numerous studies have reported an association between midwife-led care and a reduction in the incidence of potential obstetric over-interventions, along with increased maternal satisfaction during delivery [4,5,6]. Furthermore, this approach has demonstrated potential cost effectiveness, particularly in low-risk pregnancies [7,8].

It is important to note that most women who undergo vaginal births experience some form of perineal trauma. Prevention of perineal injuries is a priority for midwives, as the absence of such injuries is associated with significant benefits for women, such as reduced bleeding, reduced risk of postpartum infection and reduced perineal pain. In addition, women with an intact perineum maintain better pelvic floor tone and have fewer cases of urinary and fecal incontinence than those with perineal injuries [9].

Along these lines, the scientific literature recognizes an important set of interventions aimed at preventing perineal injuries associated with the birth process. Among these strategies, the choice of maternal position during labor has been a focus of attention. Several studies have established a direct relationship between the upright or lateral position and a significant reduction in the risk of perineal injury. Notably, the lateral position has been particularly associated with a higher rate of intact perineum. This finding is consistent with WHO recommendations, which since 1985 have advocated standing or lying on the side during the first stage of labor. Furthermore, these results reinforce the importance of considering the choice of maternal position as a key strategy to mitigate the risk of perineal injury during the birth process [10].

Currently, in Spain, the presence of various factors, such as continuous fetal monitoring and high rates of epidural analgesia or drug administration (e.g., oxytocin), has been noted to predominantly result in parturients maintaining a dorsal position during the evolution of the labor process [11].

During the birth process, postural diversification should be promoted, especially during the second stage of labor, that is, from full cervical dilatation to expulsion of the baby. The duration of this phase varies, although it tends to be longer in nulliparous women than in multiparous women [12]. Prolongation of this phase is associated with an increased risk of maternal and fetal complications [13]. Thus, postural diversification is used as a protective factor against birth canal injuries, such as episiotomies or tears, linked to short- and long-term health consequences. In the short term, these injuries are associated with low Apgar score, increased pain and hemorrhage [14], and limited maternal mobility and performance of maternal function [15]. On the other hand, in the long term, risks persist such as chronic pain, urinary or rectal incontinence [16], dyspareunia, and alterations in quality of life, among others [10].

In this context, it is crucial to explore the impact of autonomy in birth positions on perineal outcomes. Therefore, the main objective of this study is to examine the correlation between maternal positions during the second stage of labor and perineal outcomes in eutocic births attended by midwives and/or midwifery residents in a public hospital in northern Spain in 2018.

As part of our research, as a secondary objective, we will examine the incidence of perineal tears and episiotomies in eutocic births attended by midwives and midwifery residents. In addition, we will investigate the frequency with which vertical and horizontal positions are adopted during the birth process.

## 2. Materials and Methods

### 2.1. Study Design

A descriptive cross-sectional observational study was designed for eutocic deliveries attended by midwives or midwifery residents in the Labor and Delivery Service of a public hospital in northern Spain during the year 2018. Conducted within a hospital setting, it received approval from the Autonomous Research Ethics Committee of Galicia (Registration Code 2018/159). The design and implementation adhered to the Helsinki Declaration (World Medical Association, 2013) and the European Union Good Clinical Practice Directive (Directive 2005/28/EC).

### 2.2. Sample

The study population included all women who gave birth in a public hospital in northwestern Spain in 2018 (from 1 January to 31 December) who met the inclusion criteria and had none of the exclusion criteria. A non-probability circumstantial sampling method was used. Thus, women were considered eligible whose deliveries were eutocic, attended exclusively by midwives or midwifery residents between 37 and 42 weeks of gestation, with a correct recording of delivery positions during the expulsion phase, and who gave their informed consent to participate after being informed about the study. In contrast, women who were assisted by an obstetrician or obstetric resident during labor, who did not meet the pre-established inclusion criteria, and/or who did not give their consent to participate were excluded. Based on the previously described criteria, data were collected in 2018 for 367 eutocic deliveries out of a total of 692 eutocic term deliveries.

### 2.3. Procedure

At week 33, women were informed about the study during a consultation with their midwife, and an informed consent document was provided for their consideration. Upon obtaining consent from participants, the data collection process commenced. This process included the extraction of information from the women’s medical records and concluded with the comprehensive collection of data pertaining to labor, both during and at the conclusion of delivery.

### 2.4. Variables

The variables considered in this study covered socio-demographic and obstetric-gynecological aspects essential for understanding the birth process and its impact. First, socio-demographic data such as maternal age, gestational age, and parity (categorized into multiparous and primiparous) were included. In the obstetric-gynecological field, data extracted from the clinical history and the maternal positions during the third stage of labor and its final period were evaluated. The positions were grouped according to the freedom of movement they allowed, distinguishing between vertical positions (standing, hands and knees, squatting, kneeling, sitting, and delivery chair) and horizontal positions (supine decubitus, right lateral reclining, left lateral reclining, Fowler/semi-Fowler, flower with plantar support, and lithotomy), as well as those that facilitated freedom of movement and those that did not (childbirth bath).

Concerning the labor process, variables such as the presence or absence of episiotomy, the existence of tears with or without episiotomy (specifying tear grade I/II/III), and the type of analgesia administered (with/without epidural) were considered. Additionally, data related to the newborn (NB) were collected, including gender (male/female), birth weight (in grams), Apgar Test results at one and five minutes, and pH value.

### 2.5. Statistical Analysis

All analyses were performed using the statistical software SPSS 19.0 and R 3.3.2. An exhaustive descriptive analysis was conducted, presenting absolute and relative frequencies for categorical variables. For continuous variables, a normality test was performed, and means and standard deviations or medians and interquartile ranges were presented in case of rejecting normality. The descriptive analysis was complemented by graphical analysis using box plots, bar charts, or histograms. As a descriptive study, the obtained results, whether means or percentages, were directly compared within each group of interest.

## 3. Results

### 3.1. Global Description of the Sample

A total of 367 eutocic deliveries attended by midwives or midwifery residents in a public hospital in northern Spain during the year 2018 were included in the study. The mothers had a mean age of 32.5 ± 5.8 years. The mean gestational age at delivery was 40 weeks (38^+6^–40^+6^). Neonatal outcomes were satisfactory, with a mean fetal weight of 3294 g; Apgar scores at one and five minutes greater than 7 points in 98.6% and 98.2% of cases, respectively; and a mean cord pH of 7.3 ± 0.1. The remaining results of the sample can be observed in Table 1.

Regarding the longitudinal maternal axis, we observed that in 94.3% (346) of the births, the women in our study chose horizontal positions, while vertical positions were chosen in 5.7% (21) of the cases (Table 2).

In this context, the possibility of sacral retroversion in this final part of the expulsion was evidenced in the sample: 262 cases (71.7%) chose positions that did not allow sacral mobility, and 105 cases (28.3%) chose positions that did allow such mobility for fetal delivery.

Regarding the modification of posture, in our sample, changes in posture were observed in 274 births (74.7%), with no change in 93 of them (25.3%) (Table 3).

It can be seen that 67.5% of the women changed their position between one and three times during labor. On the other hand, 25.3% of the women did not change their position during labor.

### 3.2. Perineal Results

The perineal results in relation to posture are shown below. In cases where changes in posture occurred, the final position assumed was taken into consideration. This was done, excluding those deliveries with episiotomy, in order to objectively analyze those cases of expulsion without perineal injury.

In relation to the possibility of sacral retropulsion, a higher incidence of intact perineum and a lower incidence of grade III perineal injury (*p* = 0.611) was observed in postures that allowed sacral mobility (Table 4).

On the other hand, and continuing with the analysis of the possibility of maintaining a posture that allowed sacral mobility, it was noted that episiotomy was more frequent in positions where sacral retroversion was not possible. Almost in 100% of the sample (all cases except one), episiotomies were performed in positions that did not allow sacral retroversion (*p* < 0.001).

Thus, the analysis of perineal injuries in relation to the final postures of the expulsion revealed that, excluding positions with n < 3, the lithotomy position showed the lowest rate of intact perineum. In addition, it was also the position where the highest number of grade III tears were recorded, constituting 37.5% of the total sample. On the other hand, the lateral decubitus position, regardless of the side, showed grade III tears in 1.3% of births, with an intact perineum rate of 33.8%. In the case of the lithotomy position, a rate of 19.3% of intact perineum was observed.

Regarding the analysis of grade III tears, these injuries were not evident in the sitting, standing, squatting, or birthing chair positions, obtaining a rate of intact perineum of 26.31%.

The postures that showed a higher rate of perineal protection were the supine and semi-Fowler/Fowler positions with a rate of intact perineum of 34.3%. On the other hand, in these positions, the highest rate of grade III tears (6.1%) was also observed, with only 67 cases accounting for 50% of the total sample (4 cases). Finally, no grade III tears were observed in the semi-Fowler position with plantar support (footprints), despite being the position with the highest number of cases (Table 5).

Analyzing the number of position changes in relation to perineal outcomes, it was observed that 75% of third-degree tears occurred in births where there was either no change in position or only a single change occurred (Table 6).

Table 7 presents the relationship between perineal injury and other variables of interest. A statistically significant relationship was observed between primiparity and perineal injury, with an intact perineum rate of 37.8% for multiparas (*p* = 0.002). Also, statistical differences were observed based on the newborn’s weight, noting that an increase in weight was associated with an increase in the severity of the tear (*p* = 0.045).

Regarding maternal age, it was observed that a younger age of the pregnant woman acted as a perineal protective factor, as there were significant differences between maternal age and perineal injury (*p* < 0.001).

### 3.3. Episiotomy

The episiotomy rate in the sample was 15.26%, with 56 cases in total. The analysis regarding the use of episiotomy in relation to the chosen position for the fetal head’s delivery showed statistically significant differences. Thus, lithotomy constituted the position with the highest rate of recorded episiotomies, followed by the Fowler/semi-Fowler position with plantar support (footprints), with this technique observed in 19.9% of the births attended in this position (Table 8).

The study of other variables in relation to the practice of episiotomy showed that it was significantly more frequent in horizontal positions (*p* < 0.05). In this line, multiparous women obtained an episiotomy rate of 6.9%, compared to 22.8% in primiparous women (*p* < 0.001). Similarly, the newborn’s weight significantly and negatively influenced the practice of this technique, with a statistically significant relationship observed between the increase in the baby’s weight and the performance of this practice. Finally, regarding the venous pH of the umbilical cord, a lower pH value was associated with a significantly higher practice (*p* = 0.06) of episiotomy (Table 9).

### 3.4. Analgesia and Other Variables

In the study of the need for analgesia, significant differences were observed between the number of position changes and the use of epidural. No statistically significant differences were observed between tears, episiotomies, maternal age, and the type of analgesia used (Table 10).

## 4. Discussion

In our endeavor to delve into the impact of women’s autonomy in decisions related to birthing positions and its consequent influence on the process, our work focused on determining the relationship between maternal positions during the expulsive phase and perineal outcomes in eutocic births attended by midwives in a public hospital in northern Spain in the year 2018.

The low incidence of tears, with 29.4% of cases classified as grade I, 30.8% as grade II, and a minimal 2.2% as grade III, suggests effective management during the expulsive phase. These findings contrasted with tear rates observed in a similar study [11], where higher rates were recorded, especially for grade I tears. It is noteworthy that the neonatal characteristics in our study, such as the average weight of 3294 g and high APGAR scores at one and five minutes, reflected a favorable health status of the newborns.

Regarding the use of epidural analgesia, the frequency of 71.9% in this study was consistent with the rate reported in another similar study [11].

Concerning changes in birthing position, over half of the sample altered their position, ranging from a minimum of once to a maximum of six times, data similar to other previous research [17]. Among women who did not change their position, the reason for this choice is unknown; however, it may be related to the use of epidural analgesia, among other factors, as it can lead to leg weakness. Ultimately, providing postural autonomy to women during childbirth, whenever clinically feasible, may be linked to greater control over the situation, leading to relaxation and comfort until its completion [18]. Likewise, it could also be associated with facilitating the rotation and descent of the newborn’s head, reducing the duration of the expulsion phase with optimal maternal and neonatal outcomes [10,19].

According to the latest data published by the Ministry of Health of the Spanish government in 2018, the national episiotomy rate in vaginal births was 27.5% [20]. The episiotomy rate in eutocic births attended by midwives or midwifery residents in the hospital under study was 15.26%, well below the Spanish average but still slightly above the 15% that the Ministry of Health, Consumption, and Social Welfare established as a quality standard. However, studies conducted in populations similar to ours in Spain still reported episiotomy rates exceeding 20%.

A multicenter clinical trial conducted in 2019 observed a 10% rate of perineal intactness in multiparous women, emphasizing the need for a restrictive episiotomy approach to childbirth to reduce perineal trauma. These findings were consistent with several studies in which primiparity was identified as one of the main risk factors associated with episiotomy [21,22,23,24,25,26]. In addition, factors such as fetal weight greater than 4000 g and vacuum extraction have been found to be independent risk factors for both second-degree perineal tears and obstetric anal sphincter injury [21]. Other studies have also identified influential factors for the performance of episiotomy in primiparous women, such as maternal age, maternal body mass index, presence of analgesia, duration of the second stage, estimated birth weight, and perineal stress [22,23,24,25,26]. These findings led us to ask what characteristics are determinant for a higher rate of episiotomy in primiparous women and whether these characteristics justify the difference in numbers compared to multiparous women. As we continue to explore these questions, it is crucial that we continue to focus on strategies that minimize perineal trauma and improve outcomes for all women.

Through our study, a positive association between maternal age and the likelihood of perineal injury was evident. These results contrasted with other studies [27,28].

A study similar to ours conducted in Spain in 2015 [29] did not find an association between fetal weight and the practice of episiotomy. This conclusion contrasted with our results, as a statistically significant relationship between baby weight and episiotomy performance was observed.

Our study also analyzed the practice of episiotomy in relation to maternal birthing positions. This aspect is underexplored in the majority of studies that analyze the determinants of episiotomy practice. In this regard, we have demonstrated that episiotomy is more common in horizontal positions than in vertical ones, particularly in the lithotomy position and the semi-Fowler position with plantar support (heels). This finding is consistent with other consulted studies [27,29,30,31].

The likelihood of having a perineal injury-free birth increases in those births attended by midwives or midwifery residents [31]. Our findings indicated a significant association between sacral mobility and perineal injury during childbirth. Positions that facilitate sacral retroversion were linked to a higher incidence of intact perineum and a reduction in grade III perineal injuries. We also observed that almost all episiotomies were performed in positions that limited sacral mobility. In line with the existing literature, sacral mobility and vertical positions during childbirth have shown substantial benefits, reducing the duration of the expulsion phase [17,32]. This finding entails significant benefits for both the mother and the newborn.

The detailed exploration of this relationship not only seeks to contribute to the understanding of obstetric practices but also advocates for a woman-centered approach that respects her autonomy and preferences during the birthing process.

The variability in maternal positions throughout pregnancy was managed by selecting the last position assumed during the expulsion phase as the analysis criterion. However, the complexity of these data, stemming from individual diversity in postural preferences, could introduce some uncertainty in interpreting the results. The omission of the specific duration of the second stage of labor is a notable limitation in our analysis, given that the duration of this period is a crucial factor in the childbirth experience. Regarding perineal injuries of grade >3, their non-inclusion in the results is due to the absence of data, as they did not occur. This lack of detailed information represents an opportunity for future research focused on the prevention and management of perineal complications.

The limitation of conducting the study in a single hospital may affect the generalization of the results to other populations. However, this choice was based on the need to maintain homogeneity in procedures and clinical practices, allowing for a more specific and controlled analysis. Despite the mentioned limitations, this descriptive cross-sectional study represents a significant starting point in a line of research that has been sparsely explored to date. The absence of substantial previous studies in this field underscores the importance of this work as a catalyst for future research that can deepen and broaden the understanding of the analyzed obstetric factors.

In addition to the observed benefits in terms of postural autonomy and the reduction of tears and episiotomies by favoring horizontal postures during labor, there is a recognized need for a more detailed assessment of pelvic biomechanical parameters. This aspect emerges as a crucial avenue for future research that could offer a more comprehensive understanding of the relationship between maternal postures and pelvic biomechanics [33,34,35,36]. Exploring these parameters could illuminate how specific postures impact the pelvis, thus contributing to the formulation of precise recommendations for optimizing the birthing experience. Future studies are encouraged to conduct detailed biomechanical analyses, allowing a more accurate assessment of stresses and strains on the pelvis during different postures. This, in turn, would provide a more robust foundation for clinical practices and obstetric interventions.

## 5. Conclusions

In conclusion, our findings highlight a significant association between the lithotomy position and a higher incidence of episiotomy, as well as a correlation between primiparity and perineal injury. These results underscore the importance of precise clinical communication when advising women on birthing positions, empowering them to make informed decisions. Careful consideration of individual factors, such as prior birthing experience, is crucial for tailoring guidance and reducing the incidence of perineal injury, thereby contributing to a more optimal clinical management of childbirth.

## Figures and Tables

**Table 1 healthcare-12-00441-t001:** Characteristics of the study sample.

Variables	Parturition n (%)
Parity	
Multiparous	174 (47.4%)
Primiparous	193 (52.6%)
Episiotomy	
No	311 (84.7%)
Yes	56 (15.3%)
Perineal tear (including episiotomy cases)	
I	108 (29.4%)
II	113 (30.8%)
III	8 (2.2%)
None	138 (37.6%)
Perineal tear (in cases of no episiotomy)	
I	108 (33.4%)
II	113 (35.0%)
III	8 (2.5%)
None	94 (29.1%)
Analgesia	
Epidural	263 (71.9%)
No Epidural	103 (28.1%)
Newborn sex	
Woman	203 (55.3%)
Man	164 (44.7%)

Values in absolute cases and percentages in parentheses (categorical variables) or medians and interquartile ranges (continuous variables).

**Table 2 healthcare-12-00441-t002:** Distribution of positions selected by women in the study for the fetal head’s exit.

Position Type	Posture	Parturition n (%)
Horizontal		
	Supine position	6 (1.6%)
	Rigth side recline	34 (9.3%)
	Left side recline	46 (12.5%)
	Fowler/semi-Fowler	65 (17.7%)
	Flower with plantar support	136 (37.1%)
	Lithotomy	44 (12.0%)
Vertical		
	Standing	2 (0.5%)
	Hands and knees	15 (4.1%)
	Squatting	6 (1.6%)
	Kneeling	1 (0.25%)
	Seating	2 (0.5%)
	Childbirth chair	9 (2.4%)
Others		
	Childbirth bath	1 (0.3%)

**Table 3 healthcare-12-00441-t003:** Distribution of the absence or presence of posture changes during the final part of the expulsion in relation to allowing or not allowing mobility for fetal delivery.

Changes in Posture	Parturition n (%)
Number of changes	
0	93 (25.3%)
1	120 (32.7%)
2	86 (23.4%)
3	42 (11.4%)
4	19 (5.2%)
5	2 (0.5%)
6	5 (1.5%)
Change	
No	93 (25.3%)
Yes	274 (74.7%)

**Table 4 healthcare-12-00441-t004:** Distribution and *p*-value of sacral postures according to the possibility of movement in relation to the degree of discomfort.

Free Sacrum Posture	TOTAL	No Tear	IG	IIG	IIIG	*p*-Value
No	218	61 (27.9)	74 (33.8)	76 (35.1)	7 (3.2)	0.611
Yes	105	33 (31.7)	34 (32.7)	37 (34.6)	1 (1.0)	

Values in absolute cases and percentages in brackets. *p*-value calculated with the Chi-square test. IG: grade I tear; IIG: grade II tear; IIIG: grade III tear.

**Table 5 healthcare-12-00441-t005:** Distribution and *p*-value of fetal skull position in relation to the observed tear grade.

Posture	TOTAL	No Tear	IG	IIG	IIIG	*p*-Value
Posture	1	0 (0.0%)	1 (100.0%)	0 (0.0%)	0 (0.0%)	0.344
Childbirth bath	2	0 (0.0%)	1 (50.0%)	1 (50.0%)	0 (0.0%)	
Standing	15	4 (26.6%)	4 (26.6%)	7 (46.8%)	0 (0.0%)	
Hands and knees	6	2 (33.3%)	2 (33.3%)	2 (33.3%)	0 (0.0%)	
Squatting	6	3 (50.0%)	1 (16.7%)	2 (33.3%)	0 (0.0%)	
Supine position	34	13 (38.2%)	14 (41.2%)	7 (20.6%)	0 (0.0%)	
Rigth side recline	46	14 (30.4%)	12 (26.1%)	19 (41.3%)	1 (2.2%)	
Left side recline	61	20 (32.9%)	20 (32.8%)	17 (27.8%)	4 (6.5%)	
Fowler/semi-Fowler	114	30 (26.4%)	45 (39.4%)	39 (34.2%)	0 (0.0%)	
Flower with plantar support	26	5 (19.3%)	7 (26.9%)	11 (42.3%)	3 (11.5%)	
Lithotomy	0	0 (0.0%)	0 (0.0%)	0 (0.0%)	0 (0.0%)	
Kneeling	1	0 (0.0%)	0 (0.0%)	1 (100.0%)	0 (0.0%)	
Knees	2	1 (50.0%)	0 (0.0%)	1 (50.0%)	0 (0.0%)	
Seating	9	2 (22.3%)	1 (11.1%)	6 (66.6%)	0 (0.0%)	

Values in absolute cases and percentages in parentheses. *p*-value calculated with the Chi-square test. IG: grade I tear; IIG: grade II tear; IIIG: grade III tear.

**Table 6 healthcare-12-00441-t006:** Distribution and *p*-value of the number of changes in relation to the degree of tear.

Number of Changes	TOTAL n	No Tear	IG	IIG	IIIG	*p*-Value
0	86	30 (34.9%)	28 (32.5%)	25 (29.1%)	3 (3.5%)	0.611
1	109	36 (33.0%)	38 (34.9%)	32 (29.3%)	3 (2.8%)	
2	77	18 (23.4%)	22 (28.5%)	36 (46.8%)	1 (1.3%)	
3	27	5 (18.5%)	13 (48.1%)	9 (33.4%)	0 (0.0%)	
4	17	4 (23.5%)	4 (23.5%)	8 (47.0%)	1 (6.0%)	
5	2	0 (0.0%)	1 (50.0%)	1 (50.0%)	0 (0.0%)	
6	5	1 (20.0%)	2 (40.0%)	2 (40.0%)	0 (0.0%)	

Values in absolute cases and percentages in parentheses. *p*-value calculated with the Chi-square test. n: absolute value. IG: grade I tear; IIG: grade II tear; IIIG: grade III tear.

**Table 7 healthcare-12-00441-t007:** Distribution and *p*-value of perineal injuries and other variables of interest.

Posture	TOTAL n	No Tear	IG	IIG	IIIG	*p*-Value
Posture						0.342
Horizontal	302	89 (29.4%)	103 (34.1%)	102 (33.8%)	8 (2.7%)	
Vertical	21	5 (23.8%)	5 (23.8%)	11 (52.4%)	0 (0.0%)	
Change of position						0.387
No	86	30 (34.8%)	28 (32.8%)	25 (29.1%)	3 (3.3%)	
Yes	237	64 (27.0%)	80 (33.8%)	88 (37.1%)	5 (2.1%)	
Parity						**0.002**
Multiparous	164	62 (37.8%)	43 (26.2%)	56 (34.2%)	3 (1.8%)	
Primiparous	159	32 (20.1%)	65 (40.9%)	57 (35.8%)	5 (3.2%)	
Weight of newborn	3.24	3.2 (2.9–3.5)	3.2 (2.9–3.4)	3.2 (3.0–3.5)	3.6 (3.4–3.8)	**0.045**
Maternal age	33.0	31 (26.0–35.0)	33 (28.7–36.0)	35 (31.0–37.0)	33 (30.5–36.5)	**<0.001**

Values in absolute cases and percentages (categorical variables) or medians and interquartile ranges (continuous variables). *p*-value calculated with the Chi-square test (categorical variables) or Kruskal–Wallis test (continuous variables). IG: grade I tear; IIG: grade II tear; IIIG: grade III tear.

**Table 8 healthcare-12-00441-t008:** Distribution and *p*-value of the positions in relation to the use or non-use of episiotomy.

Positions	TOTAL	Episiotomy		*p*-Value
		No	Yes	
Childbirth bath	1	1 (100.0)	0 (0.0)	**<0.001**
Standing	2	2 (100.0)	0 (0.0)	
Hands and knees position	15	15 (100.0)	0 (0.0)	
Squatting	6	6 (100.0)	0 (0.0)	
Supine position	6	6 (100.0)	0 (0.0)	
Right side recline	34	34 (100.0)	0 (0.0)	
Left side recline	46	45 (97.8)	1 (2.2)	
Fowler/semi-Fowler	65	59 (90.7)	6 (9.3)	
Fowler with plantar support	136	109 (80.1)	27 (19.9)	
Lithotomy	44	22 (50.0)	22 (50.0)	
Kneeling	1	1 (100.0)	0 (0.0)	
Sitting position	2	2 (100.0)	0 (0.0)	
Birthing chair	9	9 (100.0)	0 (0.0)	

Values in absolute cases and percentages in parentheses. *p*-value calculated with the Chi-square test.

**Table 9 healthcare-12-00441-t009:** Distribution and *p*-value of other variables in relation to the use of episiotomy.

Variable	TOTAL n	Episiotomy		*p*-Value
		No	Yes	
Position				**0.045**
Horizontal	346	290 (83.8)	56 (16.2)	
Vertical	21	21 (100.0)	0 (0.0)	
Change of position				0.161
No	93	83 (89.2)	10 (10.8)	
Yes	274	228 (83.2)	46 (16.8)	
Parity				**<0.001**
Multiparous	174	162 (93.1)	12 (6.9)	
Primiparous	193	149 (77.2)	44 (22.8)	
Birth weight	3.24	3.2 (2.9–3.5)	3.36 (3.0–3.6)	**0.045**
Maternal age	33.0	33 (28–37)	34 (30–37)	0.445
Umbilical cord pH	7.26	7.26 (7.2–7.3)	7.23 (7.2–7.3)	0.060

Values in absolute cases and percentages (categorical variables) or medians and interquartile ranges (continuous variables). *p*-value calculated with the Chi-square test (categorical variables) or Mann–Whitney test (continuous variables).

**Table 10 healthcare-12-00441-t010:** Distribution and *p*-value of the need for epidural analgesia and other variables.

Variable	Total n	Epidural		*p*-Value
		No	Yes	
Frequency of position changes				**<0.001**
0	93	42 (43.2)	51 (54.8)	
1	120	31 (25.9)	89 (74.1)	
2	85	22 (25.9)	63 (74.1)	
3	42	4 (9.6)	38 (90.4)	
4	19	2 (10.6)	17 (89.4)	
5	2	1 (50.0)	1 (50.0)	
6	5	1 (20.0)	4 (80.0)	
Change of position				**<0.001**
No	93	42 (45.2)	51 (54.8)	
Yes	273	61 (22.4)	212 (77.6)	
Perineal tear				0.518
None	93	31 (33.4)	62 (66.6)	
IG	108	27 (25.0)	81 (75.0)	
IIG	113	37 (32.8)	76 (67.2)	
IIIG	8	2 (25.0)	6 (75.0)	
Episiotomy				0.124
No	310	92 (29.7)	218 (70.3)	
Yes	56	11 (19.7)	45 (80.3)	
Maternal age	33	34 (30–37)	33 (28–37)	0.445

Values in absolute cases and percentages (categorical variables) or medians and interquartile ranges (continuous variables). *p*-value calculated with the Chi-square test (categorical variables) or Mann-Whitney test (continuous variables). IG: grade I tear; IIG: grade II tear; IIIG: grade III tear.

## Data Availability

Data are contained within the article.

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
