# Peer review of "Analysis of Maternal Positions during the Dilation and Expulsive Phase and Their Relationship with Perineal Injuries in Eutocic Deliveries Attended by Midwives"

_healthcare, 2024, doi:10.3390/healthcare12040441_

Round 1
Reviewer 1 Report
Comments and Suggestions for Authors
Dear authors.
The study presented shows scientific interest and robustness, although it needs to be improved.
The fields that need improvement are: abstract, introduction and discussion.
Annex pdf with suggestions for improvement.

Author Response
Dear Reviewer,
Thank you for your input which has undoubtedly improved the scientific and technical quality of our manuscript.
In the attached document are the answers to your suggestions.
Kind regards.

Reviewer 2 Report
Comments and Suggestions for Authors
This study was conducted in 2018. The authors should justify this and if there have been significant changes in clinical practice in that hospital. Are midwives playing a similar role today in the hospital?
The manuscript have been written well in English. The abstract and key words sufficiently reflect on the study design and results.
Introduction: The literature search on position in labour and historical perspectives have been explored and are relevant to the study. The justification for this research is valid. Ethical approval has been obtained.
Method. The design of the study is acceptable. Two positions appear to be applicable for the delivery process. Variations in assuming the position at the time of delivery requires further elaboration. The title is of the study refers to the expulsive phase - does this refer to late second stage . Both pelvic and perineal phase of delivery are often discussed as part of the late second stage of labour. Perhaps some reference to this would be good.
There have been instances where patients change their position at time to delivery. This would impact on the cause-effect relationship in causing perineal injury. The authors have discussed the limitations; but this should be more explicitly mentioned.
The inclusion and exclusion criteria are highlighted. That is good.
The authors should state clearly that the final position assumed were taken to assess if there was an association with perineal injury.
Results
The tables are displayed and titled appropriately.
As a distinction is made about vertical and horizontal positions, Table 2 could show this categorization (i) all those positions placed under horizontal and (ii) those assuming vertical position. How are those in birthing chairs and delivery in bath be categorized? Some clarification why lithotomy position was assumed is required as such a position is often assumed when instrumental deliveries are conducted.
Discussion.
Some cross reference to other studies have been mentioned. The limitations of the study should include the complexity of the data as patients varied in assuming the same position throughout pregnancy. The duration of second stage is not factored in and the reasons for > 3rd degree perineal tears have not been reported. Guarding of the perineum, massage of the perineum and skills of the midwife play a role apart from maternal factors like late admission to delivery suite, maternal obesity and large fetus ( though the mean size of the fetuses were within normal range.
In concluding the discussion, suggestions for further refining and perhaps assessment of biomechanical parameters of the pelvis may be worthy of note.
References.
Adequate number are seen but all references should in English language.
Comments on the Quality of English Language
Well written except for references.
Author Response

(The authors gave the same response as above.)

Round 2
Reviewer 2 Report
Comments and Suggestions for Authors
I satisfied with the responses from the authors. They have made appropriate changes to the manuscript.